# Intra-abdominal hematomas and identifiable risk factors in patients receiving subcutaneous enoxaparin

Stephanie Lager *, Gautam Balakrishnan, Riley Eiberger

Mosaic Life Care, Saint Joseph, MO, United States of America

* stephanie.lager@mymlc.com

## Abstract

### Background

Low molecular weight heparin has proven to be safe and effective but is not without potential risks such as spontaneous bleeding in the abdominal cavity. There is limited evidence evaluating the true incidence of this potential risk and the available literature is primarily via case reports.

### Case summary

The purpose of this study was to identify the incidence and risk factors associated with enoxaparin use (prophylaxis or treatment) abdominal hematomas in a 350-bed community hospital during an 8-month time period. A total of 44 patients were identified as clinically significant bleeds receiving enoxaparin treatment or prophylactic therapy. Ultimately, 25 patients were excluded from the analysis due to an external cause of the abdominal hematoma or a temporal mismatch in enoxaparin administration and hematoma formation. After exclusion, there were a total of 19 patients that were assessed for the risk factors such as age, gender, renal function, and weight. After evaluation of risks, over half of the patients developing a clinically significant bleed were considered elderly (>65 years of age) and impaired renal function with a creatinine clearance of 60ml/min or less.

### Conclusion

Patients at risk for an enoxaparin associated hematoma include female patients with a CrCl <60ml/min and/or BMI >30 kg/m2 receiving enoxaparin treatment dosing.

## Introduction

Low molecular weight heparin (enoxaparin) safety and efficacy has been well documented in the medical literature and is considered an alternative to unfractionated heparin for prophylaxis and/or treatment of deep vein thrombosis, pulmonary embolism, atrial fibrillation, and

**Data Availability Statement:** All relevant data are within the manuscript and its supporting information files.

**Funding:** The author(s) received no specific funding for this work.

**Competing interests:** The authors have declared that no competing interests exist.

acute coronary syndrome due to its low administration frequency, decreased need for monitoring, and improved safety profile (decreased risk of thrombocytopenia) [1, 2]. Despite the safety profile of low molecular weight heparin products, case reports of spontaneous clinically significant bleeding in the abdominal cavity such as rectus sheath hematomas complicating patient care have been reported in the medical literature.

In 2001, Dabney et al. [3] reported on two cases of retroperitoneal bleeding secondary to enoxaparin that led to abdominal compartment syndrome with noting factors of advanced age and renal impairment that may predispose patients to such a risk where unfractionated heparin may be a safer alternative with increased monitoring utilizing the aPTT.

In 2006, Lee et al. [4] reports an incidence of retroperitoneal hematoma during enoxaparin treatment of pulmonary embolism. The patient was an elderly female on treatment dosing (1 mg/kg) of enoxaparin in addition to warfarin. While the case report did not specify the patient's creatinine clearance, there was notable renal dysfunction present as evidenced by an elevated serum creatinine. Authors concluded that as enoxaparin therapy becomes more frequently used, the incidence of adverse events such as retroperitoneal hematomas will continue to increase.

Quartey et al. reported a case [5] in 2011 with a 77-year-old white male patient with mild renal insufficiency who developed a massive spontaneous retroperitoneal hemorrhage secondary to enoxaparin that led to compartment syndrome that required surgical intervention. The author also conducted a thorough review of the current literature and identified fewer than 10 cases of spontaneous retroperitoneal hematomas and concluded that patients with advanced age (greater than 60 years), renal impairment and concomitant medications that increased bleeding (aspirin, clopidogrel) are at an increased risk for massive bleeding events and caution should be advised in using enoxaparin in patients with these risk factors.

A fatal case report [6] was reported by Salemis et al. in 2014 in a 75-year-old female patient with no bleeding history and normal renal function presenting with worsening abdominal pain over the last 12 hours accompanied by tachycardia. Of note, the patient was previously on warfarin which was discontinued and bridged with enoxaparin to prepare for an upcoming surgery. The patient required emergent surgery for stabilization; however, later died despite heroic resuscitative efforts. Salemis concluded that despite this being a rare complication with a 5% incidence and with reports occurring within 5 days of therapy, similar risk factors were identified such as advanced age, impaired renal function and treatment doses of enoxparin.

Finally, in 2020 Mendes et al. [7] documented a series of 5 case reports of rectus sheath hematomas in a single hospital setting within a four-month period. Researchers evaluated each of the patients for identifiable risk factors that led to the hematoma development. In analyzing the patients, 3 out of 5 patients were female and all 5 patients were greater than 65 years of age. Four of the five patients were on treatment dosing of enoxaparin with only one patient receiving prophylactic dosing with all doses adjusted for renal function. Ultimately, the author concludes the following identified risk factors of age greater than 65, female sex, kidney dysfunction, and concomitant use of additional blood thinners or antiplatelet medications that would pose an increased risk of spontaneous bleeding associated with enoxaparin and advises clinicians to identify patients with such an adverse event such as those with abdominal pain, mass or distension, a decrease in hematocrit, hypotension, and/or tachycardia.

Due to the nature of the available literature (case reports), the true incidence of abdominal hematomas associated with enoxaparin is unknown. The purpose of this study was to identify the incidence and risk factors associated with enoxaparin use abdominal hematomas in a 350-bed community hospital.

## Materials and methods

The retrospective chart review was approved by the Mosaic Institutional Review Board (Approval #122121 and approved on 12/21/21) and conducted on all patients aged 35 years or older and admitted to a 350-bed hospital from March 1, 2020 to November 1st, 2021 that received enoxaparin prophylaxis or treatment dosing with a CT illustrating clinically significant bleeding. A natural language processing tool with Nuance Healthcare solutions-mPower clinical solutions was used with key words of hemorrhage and or hematoma to identify those patients with clinically significant bleeding. Patient data was accessed from 12/22/21 to 2/14/22 for research purposes only. Each CT scan was initially read by a staff radiologist with documentation of the bleed in the electronic medical record. At study enrollment, each CT was again reviewed personally by a single physician also, the author, to confirm clinically significant bleeding to ensure relevant cases were identified. Patients excluded included trauma or those with a CT performed in the Emergency room as to identify inpatients only for inclusion in the study.

Patient demographics were collected in addition to clinical data evaluating the patient's renal function utilizing the Cockcroft Gault equation, other medications that may contribute to bleeding, concurrent use of nephrotoxic medications, body mass index (BMI), and the use of prophylactic versus treatment enoxaparin dosing methods. Patient transfusion requirements 3 days prior and 3 days after the bleeding event and the administration of antidotes was also collected. In addition, outcomes such as the need for interventional radiology and survival were evaluated.

Patients were excluded if they were pregnant, a prisoner, or if the bleeding event was associated with another cause and not associated with administration of enoxaparin.

## Results and discussion

A total of 44 patients were identified as clinically significant bleeds receiving enoxaparin treatment or prophylactic therapy. Ultimately, 25 patients were excluded from the analysis due to an external cause of the abdominal hematoma or a temporal mismatch in enoxaparin administration and hematoma formation. After exclusion, there were a total of 19 patients that were assessed for the aforementioned risk factors. See Table 1

When evaluating the risk factors, over half of the patients developing a clinically significant bleed were considered elderly (>65 years of age). Regarding renal function, there were 5 (26.3%) patients with a creatinine clearance (CrCl) less than or equal to 30 mL/min, 9 (47.4%) patients with a CrCl between 30 mL/min and 60 mL/min, and 4 (21.1%) patients with a CrCl greater than or equal to 60 mL/min.

When BMI was assessed, there were no patients less than or equal to 18.5 kg/m$^2$, nine patients (47.4%) with a BMI of 18.5–27 kg/m$^2$, one patient (5.3%) with a BMI of 27–30 kg/m$^2$, and 8 patients (42.1%) greater than or equal to 30 kg/m$^2$.

There were 2 patients (10.5%) who received a blood transfusion in the 3 days prior to their hematoma and 17 patients (89.5%) who did not receive a blood transfusion in the 3 days prior to their hematoma. However, after the bleeding event, 11 patients (57.9%) received a blood transfusion in the 3 days following their hematoma while 8 patients (42.1%) patients didn't receive any blood products.

Lastly, there were 7 patients (36.8%) who required embolization of their hematoma by interventional radiology while the remaining 12 patients (63.2%) did not require embolization and only one patient. (5.3%) received an antidote (protamine) for their enoxaparin therapy. Of the 19 patients noted to have an abdominal hematoma, 15 patients (78.9%) survived while 4 patients (21.1%) expired.

**Table 1. Identifiable risk factors for intra-abdominal hematoma.**

| Age | |
|---|---|
| Greater than or equal to 65 years of age | 12 patients (63.2%) |
| Less than 65 years of age | 7 patients (36.8%) |
| **Body Mass Index** | |
| $\leq 18.5$ kg/m$^2$ | 1 patient (5.3%) |
| 18.5–27 kg/m$^2$ | 9 patients (47.3%) |
| 27–30 kg/m$^2$ | 1 patient (5.3%) |
| $\geq 30$ kg/m$^2$ | 8 patients (42.1%) |
| **Gender** | |
| Male | 8 patients (42.1%) |
| Female | 11 patients (57.9%) |
| Creatinine Clearance | |
| $\leq 30$ mL/min | 5 patients (26.35%) |
| 30 mL/min-60 mL/min | 9 patients (47.3%) |
| $\geq 60$ mL/min | 5 patients (26.35%) |
| **Nephrotoxic Medications** | |
| Yes | 8 patients (42.1%) |
| No | 11 patients (57.9%) |
| Other anticoagulants or antiplatelets | |
| Yes | 8 patients (42.1%) |
| No | 11 patients (57.9%) |
| **Enoxaparin Dosing: Treatment vs Prophylaxis** | |
| Treatment | 11 patients (57.9%) |
| Prophylaxis | 5 patients (26.35%) |
| Both | 5 patients (15.75%) |
| **Enoxaparin Dosing Appropriate Based on Package Insert** | |
| Yes | 16 patients (84.2%) |
| No | 3 patients (15.8%) |
| **Survival** | |
| Yes | 15 patients (78.9%) |
| No | 4 patients (21.1%) |
| **Embolization** | |
| Yes | 7 patients (36.8%) |
| No | 13 patients (36.2%) |
| **Antidote** | |
| Yes | 1 patient (5.3%) |
| No | 18 patients (94.7% |
| **Blood Transfusion 3 days Prior to Hematoma Identification** | |
| Yes | 2 patients (10.5%) |
| No | 17 patients (89.5%) |
| **Blood Transfusion 3 days following hematoma identification** | |
| Yes | 11 patients (57.9%) |
| No | 8 patients (42.1%) |

"We identified patients from March 2020 during the start of the pandemic till November 2020 with three of our nineteen patients positive for COVID-19, eight patients tested but with a negative result and 8 patients not tested during his or her hospital admission With COVID-19 infected patients who are critically ill there is a high risk of prothrombotic, and micro-

micro thrombotic events described with supplemental anticoagulation therapy that is routinely used. Full dose anticoagulation with heparin in high-risk patients without evidence of thrombotic disease did not have a significant mortality benefit on clinical trials [8]. Spontaneous retroperitoneal bleeds secondary to anticoagulant therapy is a known complication but not very well reported. With COVID-19 patients receiving intermediate or therapeutic dose anticoagulation, Interleukin– 6 blockade, steroids, all increases the risk for major bleeding. Along with the fact that predominantly inpatient COVID-19 pneumonia with elevated levels of D-dimer, ferritin with a very pro inflammatory disease overall the risk of bleeding could be high. It is unclear if there is a reason enoxaparin should have a high risk of bleeding compared to heparin. Subgroup analysis [9] in the past with efficacy and safety of enoxaparin versus unfractionated heparin is actually showed major bleeding was reduced by 34% compared with unfractionated heparin. Low molecular weight heparin administered subcutaneously was actually found to be more effective than the use of dose adjusted intravenous unfractionated heparin for preventing DVT/PE [10].

In an eight-month time period, this study reported a total of 19 patients with clinically significant bleeding secondary to enoxaparin administration within one institution. Due to the decreased incidence of approximately 5% as reported in Salemis et al. and fewer than 20 cases reported in the literature over the past 20 years, this adverse event and its true incidence is likely underreported. There is also lack of evidence about the risks of developing a spontaneous bleeding or hematoma with the use of enoxaparin and the literature that does exists is in small case reports with few patients. This study had similar findings to Mendes et al where a majority of patients were female and had impaired renal function defined as a CrCl <60ml/min.

Another potential risk factor identified that was not included in other case reports was body mass index. In this study, most patients were within normal limits for weight; however, those that were considered morbidly obese with a BMI of 30 kg/m2 or greater may be at an increased risk for the development of spontaneous clinically significant bleeding. Evidence is limited on medications and dosing in the obese population; however, based on our findings, caution should be exercised when dosing enoxaparin in this patient population with simultaneous renal dysfunction.

This study was not without limitations which included the retrospective case report design with the full clinical condition of each patient being unknown and therefore the possibility of bleeding influenced by other factors that are not reported. However, clinicians should be familiar with this adverse event associated with enoxaparin and watch for clinical signs that would lead to further workup and imaging to identify a possible hematoma.

## Conclusion

Patients at risk for an enoxaparin associated hematoma include female patients with a CrCl <60ml/min and/or BMI >30 kg/m2 receiving treatment dosing.

## Supporting information

**S1 File. Rectus sheath hematoma excel sheet.**
(ZIP)

## Acknowledgments

We would like to acknowledge Sara Mills, Pharm.D. for her assistance proofreading and review of the final manuscript.

## Author Contributions

**Conceptualization:** Gautam Balakrishnan.

**Data curation:** Riley Eiberger.

**Formal analysis:** Stephanie Lager.

**Investigation:** Gautam Balakrishnan, Riley Eiberger.

**Methodology:** Gautam Balakrishnan.

**Project administration:** Stephanie Lager.

**Software:** Gautam Balakrishnan.

**Supervision:** Stephanie Lager, Gautam Balakrishnan.

**Writing – original draft:** Stephanie Lager, Riley Eiberger.

**Writing – review & editing:** Stephanie Lager, Gautam Balakrishnan, Riley Eiberger.

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
