## [Decision Letter · Decision Letter 0]

13 Feb 2024

PONE-D-23-42266Intra-abdominal Hematomas and Identifiable Risk Factors in Patients Receiving Subcutaneous EnoxaparinPLOS ONE

Dear Dr. Lager,

Thank you for submitting your manuscript to PLOS ONE. After careful consideration, we feel that it has merit but does not fully meet PLOS ONE’s publication criteria as it currently stands. Therefore, we invite you to submit a revised version of the manuscript that addresses the points raised during the review process.

We look forward to receiving your revised manuscript.

Kind regards,

Gagandeep Dhillon, MD, MBA

Academic Editor

PLOS ONE

A clean copy of the edited manuscript (uploaded as the new *manuscript* file).

4. Please include your table as part of your main manuscript and remove the individual file.

Additional Editor Comments:

Dear authors,

I hope you are doing well. Congratulation on doing a great job with the study. The information in this report was well researched, planned, and executed. The work quality is good but needs minor changes including grammar and sentence restructuring. The study has a well defined objective and the fact that it was conducted in a 350-bed community hospital makes its findings more applicable to similar healthcare settings. I was also impressed with the evaluation of various risk factors, including age, gender, renal function, and weight, as it provides a comprehensive understanding of the patient population at risk. However, the sample size of 19 patients is relative small and being a retrospective study, it is susceptible to inherent biases, and causality cannot be definitively established. Overall. it addresses a clinically relevant issue.

Please look at the comments made by the reviewers and address them. Also, I had a few questions for you.

1. “Each CT scan was reviewed personally by a single physician to confirm clinically significant bleeding to ensure relevant cases were identified.”

Is the physician one of the authors?

2. “With COVID-19 patients receiving intermediate or therapeutic dose anticoagulation, Interleukin – 6 blockade, steroids, all increases the risk for major bleeding”

Please elaborate on this

3. What was the lovenox dose used for orbidly obese with a BMI of 30 kg/m2?

I look forward to receiving your response.

Reviewers' comments:

Reviewer's Responses to Questions

**Comments to the Author**

1. Is the manuscript technically sound, and do the data support the conclusions?

Reviewer #1: Yes

Reviewer #2: Yes

2. Has the statistical analysis been performed appropriately and rigorously? 

Reviewer #1: No

Reviewer #2: Yes

3. Have the authors made all data underlying the findings in their manuscript fully available?

Reviewer #1: Yes

Reviewer #2: Yes

4. Is the manuscript presented in an intelligible fashion and written in standard English?

Reviewer #1: Yes

Reviewer #2: Yes

5. Review Comments to the Author

Reviewer #1: This study provides crucial insights into enoxaparin-associated abdominal hematomas, emphasizing risk factors such as age, renal impairment, and obesity. Despite its retrospective design and limited sample size, the findings align with existing literature, contributing valuable knowledge to guide clinicians in cautious enoxaparin dosing.

Reviewer #2: The authors did great job looking into the incidence of abdominal hematoma after enoxaparin use. This is a retrospective study, which have a few limitations and well mentioned in the discussion. It would be great to add more information about the concurrent presence or absence of COVID-19 infections in those patients included in analysis. It would appropriate to add year after November at one place. Overall, it is well written.

6. PLOS authors have the option to publish the peer review history of their article (what does this mean?). If published, this will include your full peer review and any attached files.

Reviewer #1: **Yes: **Ripudaman Munjal

Reviewer #2: No

---

## [Author Response · Author response to Decision Letter 0]

21 Mar 2024

Academic Editor 

1. Yes, the physician that again read the CT scan to confirm the previous reading is also an author. This section was modified to make it transparent to the reader (see below). 

Each CT scan was initially read by a staff radiologist with documentation of the bleed in the electronic medical record. At study enrollment, each CT was again reviewed personally by a single physician also, the author, to confirm clinically significant bleeding to ensure relevant cases were identified.

2. “With COVID-19 patients receiving intermediate or therapeutic dose anticoagulation, Interleukin – 6 blockade, steroids, all increases the risk for major bleeding”

Please elaborate on this

This section was expanded and revised a little especially in regards to the number of positive covid-19 patients we had; however, there is still a lot of unknowns in this area that do not allow us to elaborate more at this time. 

3. What was the lovenox dose used for morbidly obese with a BMI of 30 kg/m2?

The section regarding enoxaparin dosing was modified and addressed the dosing in morbidly obese patients. 

Enoxaparin prophylactic dosing consisted of 40mg SQ daily unless the patient creatinine clearance (CrCl) was less than or equal to 30ml/min at which the dosing was reduced to 30mg SQ daily. Enoxaparin treatment dosing consisted of 1mg/kg SQ twice daily unless the patient creatinine clearance was less than or equal to 30ml/min at which the dosing was reduced to 1mg/kg SQ daily. Actual body weight was used for patient dosing and Body Mass Index (BMI) was not taken into consideration with alternative dosing. 

Reviewer 1

With the limited number of cases and the financial constraints to conduct a prospective study, we conducted a retrospective review to begin to answer the question is the side effect of intra-abdominal hematomas more common than that which is published in the literature to date. 

Reviewer 2

Has the statistical analysis been performed appropriately and rigorously? 

No because as above, with the limited number of cases and the financial constraints to conduct a prospective study, we conducted a retrospective review to begin to answer the question is the side effect of intra-abdominal hematomas more common than that which is published in the literature to date and we believe our results do answer that question and can raise awareness among providers when caring for patients treated with enoxaparin.

Reviewer 3

Have the authors made all data underlying the findings in their manuscript fully available?

The patient data has been deidentified and will be sent with the original manuscript and will support the findings summarized in our research article. 

Reviewer 4

Our article was reviewed by the authors again with grammatical and formatting changes and then sent to another reviewer and researcher for feedback on proofing and editing. 

Reviewer 5

We did go back and review our patients again to determine how many were affected by the Covid-19 pandemic. Upon review, 3 of our patients were positive with 8 patients testing negative and the remaining 8 were not tested on his or her hospital admission. These details were added to the manuscript.

---

## [Editor Report · Decision Letter 1]

25 Mar 2024

Intra-abdominal Hematomas and Identifiable Risk Factors in Patients Receiving Subcutaneous Enoxaparin

PONE-D-23-42266R1

Dear Dr. Lager,

We’re pleased to inform you that your manuscript has been judged scientifically suitable for publication and will be formally accepted for publication once it meets all outstanding technical requirements.

Kind regards,

Gagandeep Dhillon, MD, MBA

Academic Editor

PLOS ONE

Additional Editor Comments (optional):

Dear authors

I wanted to take a moment to commend you on the changes you made to your article based on the recommendations I provided. Your dedication to improving the quality and effectiveness of your work is truly admirable.

The revisions you implemented demonstrate not only your receptiveness to feedback but also your ability to thoughtfully incorporate suggestions into your writing. The adjustments have significantly enhanced the overall clarity and impact of the article.
---

## [Editor Report · Acceptance letter]

2 Apr 2024

PONE-D-23-42266R1 

PLOS ONE

Dear Dr. Lager, 

I'm pleased to inform you that your manuscript has been deemed suitable for publication in PLOS ONE. Congratulations! Your manuscript is now being handed over to our production team.

Kind regards, 

on behalf of

Dr. Gagandeep Dhillon 

Academic Editor

PLOS ONE